# Thermally Remendable Polyurethane Network Cross-Linked via Reversible Diels–Alder Reaction

**DOI:** 10.3390/polym13121935

**Published:** 2021-06-10

**Authors:** Elena Platonova, Islam Chechenov, Alexander Pavlov, Vitaliy Solodilov, Egor Afanasyev, Alexey Shapagin, Alexander Polezhaev

**Affiliations:** 1Laboratory of Functional Composite Materials, Bauman Moscow State Technical University, 2nd Baumanskaya str., 5/1, 105005 Moscow, Russia; E-O-Platonova@yandex.ru (E.P.); ch-islam@bk.ru (I.C.); vital-yo@ya.ru (V.S.); 2Laboratory for Nuclear Magnetic Resonance, A.N. Nesmeyanov Institute of Organoelement Compounds, Vavilova str., 28, 119334 Moscow, Russia; pavlov@ineos.ac.ru; 3Laboratory of Reinforced Plastics, Semenov Institute of Chemical Physics, Russian Academy of Sciences, 119991 Moscow, Russia; 4Laboratory for Polymer Materials, A.N. Nesmeyanov Institute of Organoelement Compounds, Vavilova str., 28, 119334 Moscow, Russia; nambrot@yandex.ru; 5Laboratory of Structural and Morphological Investigations, Frumkin Institute of Physical Chemistry and Electrochemistry, Leninsky Prospect 31, bld.4, 119071 Moscow, Russia; shapagin@mail.ru

**Keywords:** polyurethanes, Diels–Alder reaction, crosslinked polyurethanes, recyclability

## Abstract

We prepared a series of thermally remendable and recyclable polyurethanes crosslinked via reversible furan-maleimide Diels–Alder reaction based on TDI end-caped branched Voranol 3138 terminated with difurfurylamine and 4,4′-bis(maleimido)diphenylmethane (BMI). We showed that Young modulus strongly depends on BMI content (from 8 to 250 MPa) that allows us to obtain materials of different elasticity as simple as varying BMI content. The ability of DA and retro-DA reactions between furan and maleimide to reversibly bind material components was investigated by NMR spectroscopy, differential scanning calorimetry, and recycle testing. All polymers obtained demonstrated high strengths and could be recovering without significant loss in mechanical properties for at least five reprocessing cycles.

## 1. Introduction

Global polyurethane production is more than 22 million tons (for PU foams), and most of this production will not be recycled [1,2]. The synthetic design of recyclable materials is essential for sustainability, and there is a noticeable growth in the number of papers in the field in the last decade [1]. There are two ways to increase polyurethane sustainability: developing materials based on biorenewable sources such as lignin or sugars [3,4,5,6] and developing recycling technologies [7,8]. Generally, PU may be recycled by either destructive or nondestructive methods. Linear polyurethanes can be reprocessed through conventional molding processes like other thermoplastics [9,10,11,12]. However, the polyurethane polymer family is not limited to thermoplastics, but there are many crosslinked formulations required for high solvent resistance [13,14], high strength [15,16], and abrasion resistance [17]. Crosslinked polymers cannot be reshaped or repaired and finally recycled when damaged once a crosslinked network is formed [10,18]. There are several approaches to commercial crosslinked PUs recycling: advanced chemical and thermochemical recycling, mechanical recycling, energy recovery, and product recycling, but all of them suffer from inefficiency and high cost [1,10].

The introduction of covalent bonds, which can reversibly dissociate upon extrinsic stimuli, allows to obtain mendable crosslinked materials [19] and could substantially simplify material recycling [20,21,22,23]. Reversible Diels–Alder reaction between furan and maleimide is one of the most sustainable approaches to remendable materials development [24] with the advantage of inexpensive biomass-derived furans [25,26,27] and commercially available bismalemide crosslinkers as main components. A technological thermal window required for processing is usually around rt—70 °C for bonding and 100–140 °C for the reversed process [28]. This approach is also widely used for the development of self-healing polymer systems [29,30]. It is also essential for sustainability to increase the number of carbon atoms derived from renewable sources. Furans, maleimides [31], and polyols [11,32,33,34] are bioderived and therefore are ideal candidates for this purpose. Our previous work [31] presented linear thermally remendable PUs extended via reversible DA-reaction that demonstrated excellent self-healing ability. Several reports showed [35,36,37] that branched PUs have enhanced mechanics and self-healing properties than their linear counterparts. For example, Du et al. achieved crosslinked material by reaction between secondary amino-groups of furylamine-functionalized prepolymer and isocyanate groups of 4,4-diphenylmethane diisocyanate-functionalized preform at elevated temperature [35]. Later the same authors used branched triol as a chain extender for branched PU synthesis [36]. Here we decided to increase a soft segment molecular mass compared to M = 1000 in Du at all. [35,36]), as far as elongation of the soft segment in remendable PUs to achieve a positive influence on polymer chains mobility and consequently enhances self-healing properties and potential recyclability [38].

We designed netlike remendable PU using branched glycerine-based propylene oxide and ethylene oxide block-copolymer, Voranol 3138, as the soft segment. Previously polyols of such or similar structures were used for foams [39,40] and amphiphilic materials [41] fabrication, but not for self-healing or mendable polymers. A new furan-rich component difurfurylamine, synthesized from furfural, was applied for increasing local furan content and speeding up DA-adduct formation compared to the know systems based on furfuryl alcohol or furfurylamine. We choose commercially available methylenedi-1,4-phenylene)bismaleimide (BMI) as a crosslinker. It is worthwhile to say that a significant advantage of the furan-terminated prepolymer is its longer shelf-life and moisture insensitiveness compared to its regular isocyanate-terminated counterparts [42].

## 2. Materials and Methods

### 2.1. Materials

TDI (toluene-2,4-diisocyanate, 80%), sodium borohydride (98%), BMI (1,10-(methylenedi-1,4-phenylene)bismaleimide, 95%), and BHT (2,6-di-tert-butyl-4-methylphenol, 99.0%) were purchased from Aldrich and used as received. Voranol 3138 (M_n_ = 3000) was purchased from Dow Europe GmbH and dried under vacuum at 110 °C prior to use. Hydrofuramide was prepared using a published procedure [43] from furfural and ammonia. DMF (N,N-dimethylformamide) was purchased from Acros dried over CaH_2_, and distilled prior to use.

### 2.2. Synthesis of Difurfurylamine

Difurfurylamine was prepared by a modified literature procedure [31] (Scheme 1). Hydrofuramide (50 g, 0.19 mol) was gradually added to the ice-cold suspension of sodium borohydride (excess) in methanol. The reaction mixture was stirred for an additional 0.5 h after complete hydrofuramide dissolution and then diluted with water to 1 L and extracted with CH_2_Cl_2_. The organic layer was washed with water and brine and dried with Na_2_SO_4_. Volatiles were removed by rotary evaporation, and the resulting crude product was distilled in vacuo (130 °C/16–18 mbar) to obtain almost colorless liquid (27.2 g, 55%).

### 2.3. Synthesis of PU-V0

Prepolymer PU-V0 was prepared using conventional two-step method. At the first step TDI (2.3 g, 13.4 mmol) reacted with Voranol 3138 (12.5 g, 4.2 mmol) in a 100 mL two-necked round bottom flask, equipped with magnetic stirrer and dropping funnel. The reaction was carried out at 60 °C under an argon atmosphere for 3 h. The reaction progress was monitored with IR-spectroscopy. At the second step, the reaction mixture was cooled to room temperature, diluted with DMF (3 mL), and difurfuryl amine (2.3 g, 13.0 mmol) dissolved in DMF (3 mL) was added dropwise into the isocyanate end-capped prepolymer solution in about 15 min. Reaction mixture was stirred at 60 °C under an argon atmosphere for 12 h. Yield: 21.0 g. ^1^H NMR (400 MHz, DMSO-*d*_6_, δ): 9.52 (s, 3H, –N**H**–CO–O–), 8.05 (d, *J_HH_* = 9.8 Hz, 3H, –HNC(O)N**H**–Ar), 7.61 (d, *J_HH_* = 6.9 Hz, 3H, =C**H**O– of furan ring), 7.44 (d, *J_HH_* = 6.4 Hz, 3H, –Ar), 7.26–7.12 (m,, 3H, –Ar), 7.04 (d, *J_HH_* = 5.3 Hz, 3H, –Ar), 6.41 (s, 6H, =C**H**–C**H**= of the furan ring), 6.32 (s, 6H, =C**H**–C**H**= of the furan ring), 4.54 (s, 12H, –N–C**H****_2_**–), 3.71–3.62 (m, 5H, –O–C**H_2_**–C**H**(O)–C**H_2_**O–), 3.57–3.39 (m, 112H, –C**H****_2_**– of polypropylene oxide and polyethylene oxide), 3.35–3.17 (m, 66H, –CH_2_–C**H**(CH_3_)–O–), 1.20 (s, 9H, C**H****_3_**–Ar), 1.04 (d, *J_HH_* = 5.0 Hz, –O–CH(C**H****_3_**)–CH_2_–); ^13^C NMR (101 MHz, DMSO-*d*_6_, δ): 155.58, 153.68 151.91, 143.1 138.37, 138.23, 130.32, 123.13, 118.05, 110.93, 108.59, 75.04, 72.71, 42.81, 17.43; IR (ATR, neat, cm^−1^): 3315, 2971, 2931, 2895, 2897, 1728, 1672, 1600, 1531, 1506, 1453, 1374, 1343, 1296, 1225, 1103, 1012, 924, 868, 816, 737.

Details on material characterization can be found in the Appendix A.

### 2.4. Synthesis of Crosslinked PUs

PU-V1.0. BMI (0.93 g, 2.5 mmol) and BHT (0.0057 g, 0.05 mmol, 1 %mol to the amount of prepolymer and crosslinker) as stabilizer were added to Fu_2_N-prepolymer PU-V0 (7.0 g, 1.7 mmol) in 6 mL DMF. The reaction mixture was heated to 60 °C for 3 h, then carried onto a glass surface and cured in an oven for 48 h to afford semitransparent polymer film. Polyurethanes with different prepolymer:BMI ratios (PU-V2.0 and PU-V3.0) were synthesized using the same protocol.

Details on material characterization can be found in the Appendix A.

### 2.5. Film Preparation

Polymer solution in DMF (0.12 g mL^−1^) was heated to 60 °C, poured into a glass Petri dish, and kept in an oven at 60 °C for 48 h. All films were stored at room temperature not more than a week before testing.

### 2.6. Recycling of PU-V1.0-3.0

A 50 mL of DMF was added to 10 g of cut polymeric material, and a mixture was heated to 100 °C while stirring until complete dissolution. The hot solution was poured into PTFE form and cured at 60 °C for 48 h.

### 2.7. Characterization

NMR spectra were recorded by a Bruker Avance 600 NMR Spectrometer (600.1 MHz), residual proton signal of deuterated solvent was used as reference. ATR-FTIR was performed by Nicolet iS10 spectrometer in the range of 4000 to 600 cm^−1^ on a germanium crystal. Thermogravimetric analysis (TGA) was performed by Netzsch TG 209 F1 Libra within a temperature range of 30–550 °C at a heating/cooling rate of 10 K min^−1^ under an argon atmosphere. Differential scanning calorimetry (DSC) was performed by a Netzsch DSC 204 F1 Phoenix within a temperature range of −80–300 °C at heating/cooling rates of 10 K min^−1^ (−80–160 °C at heating/cooling rates of 5 K min^−1^ for T_g_ determination) under an argon atmosphere. A sample weight of about 20 mg was used for each measurement. Thermal analysis was performed by TA Instruments TMA Q400E in the penetration mode. Specimens of 0.6 cm diameter were tested within a temperature range of −70–390 °C at heating/cooling rates of 5 °C min^−1^, at the load of 1 N, and penetration probe diameter of 2.54 mm. Transmission electron microscopy was performed by a Philips TEM-301. Samples for TEM were prepared using the replica technique. Cryo-fractured surfaces were obtained in liquid nitrogen. Samples phase structure was identified by oxygen plasma etching in 40 min on Edwards E306A evaporator. Mechanical properties (*σ_t_*, *E*, *ε*, *σ_t_* (*ε*) were determined from the uniaxial tensile strength test. Dogbone-shaped samples were cut from film using a special blade (length 15 mm, width 2 mm, see Appendix A and Appendix A in the Appendix A) and hand press. Film thickness was from 1.5 to 2.0 mm. The samples were stretched on a Zwick Roell Z100 universal testing machine at a 150 mm/min loading speed. An applied load *P* and a strain *ε* were recorded as loading diagrams. Further, the stress at break *σ_t_*, the elongation *ε*, and the elastic modulus *E* were calculated from the diagrams.

## 3. Results and Discussion

### 3.1. Synthesis and Characterization of PU-V0 and PU-V1.0-3.0

Thermally remendable PUs containing difuran moieties were prepared by a two-step polymerization process and were characterized by FTIR and ^1^H-NMR spectra (Scheme 2) using the same protocol we previously designed [31,44]. Briefly, Voranol 3138 reacted with 3 mol of TDI at 60 °C and then terminated with difurfurylamine resulted in transparent viscous liquid (PU-V0). Reaction progress was monitored with ATR-IR, and the disappearance of the isocyanate band at 2265 cm^−1^ indicated completion of the reaction between difurfurylamine and isocyanate [45]. A sample of PU-V0 and BMI in different molar ratios (1:1.5 for PU-V1.0, 1:2 for PU-V2.0, and 1:2.5 for PU-V3.0) were dissolved in DMF, cast in glass, or PTFE mold, and heated at 60 °C for 48 h in the oven yielded elastic yellow, semitransparent films. BHT (1% mol) was added to a solution to prevent maleimide polymerization. When less than 1.5 eq. of BMI was used, the curing was incomplete, and the film formed was sticky and not suitable for testing. An excess of unreacted BMI crystallizes on the film surface when more than 2.5 eq. of BMI has been used.

The FTIR spectra of the furan-terminated oligomer PU-V0, BMI, and films of crosslinked PU-V1.0-3.0 are shown in Figure 1. A decreasing of the band at 741 cm^−1^ associated with furan ring and appearance of the weak band at 1776 cm^−1^ characteristic to a furan-maleimide Diels–Alder adduct in the spectra of polymers PU-V1.0-3.0 confirmed DA-adduct formation [46,47]. However, the presence of two C=O bands stretches at 1707 cm^−1^ may be assigned to partially unreacted maleimide groups. We speculate that incomplete behavior of the DA-reaction between prepolymer and BMI may be caused by increasing viscosity. Continuous heating of the sample did not change spectra, meaning that a higher conversion of the starting material cannot be obtained with prolonged heating. That means increasing furan content in the preform structure is essential to obtained higher crosslink density. 

All polymers obtained (PU-V1.0-3.0) were not soluble in DMSO at r.t. and no leakage of BMI was detected with NMR after 12 h soaking of polymer in DMSO-d^6^ in NMR tube compared to PPG2000-based polymers that were perfectly soluble in DMSO. Prolonged heating at 100 °C was necessary to obtain a homogeneous solution required for ^1^H-NMR spectra registration. We believe that at least a partial rDA reaction occurred during this process (Figure 2). The ^1^H-NMR spectra of preheated samples showed residual furan ring signals at 7.62, 6.42, and 6.32 ppm, and at the same time, a new signal of DA adduct vinyl protons appeared at 6.52–6.26 ppm. All three polymers’ proton NMR spectra contained signals from a crosslinking agent, which may be the result of DA-bonds cleavage caused by heating upon specimen preparation. The multiplet at 4.87 ppm corresponded to protons of N–C(O)–O–C*H*_2_-fragment [41]. 

Soaking 6 g of the PU-V3.0 in 20 mL of DMF for 12 h leaded to swelling and consumption of all DMF used. Washing of the specimen with more DMF and analysis of a soluble fraction showed <10% of the mass leached to DMF mostly consisted of BMI. The above results indicate that the crosslinked PUs were successfully obtained in the mold and prolonged heating over 100 °C in a polar solvent is required to break the network.

In our previous study, we presented linear remendable PU with good processability and solubility in DMF (even at 60 °C) [31]. By contrast with linear PUs netlike PU-V1.0-3.0 were almost insoluble in common organic solvents (DMF, DMSO, acetone) at room temperature and swelled in a solvent with gel formation. Only by heating those gels to the rDA temperature > (100 °C) we obtained homogeneous solutions.

### 3.2. Thermal Properties and Reversibility of the DA Bonds

The thermal stability of PU-V1.0-3.0 was investigated with TGA under an argon atmosphere (Figure 3a). The temperature values at 5% mass loss (T5%) were 267, 279, and 285 °C for PU-V1.0-3.0, respectively. All polymer samples showed a slow weight loss in the range of 260–360 °C (19.0%, 21.8%, and 21.3% for PU-V1.0-3.0, respectively), suggesting their high thermal stability. Main weight loss occurs in the range of 370–450 °C (85.9%, 83.2%, and 75.0% for PU-V1.0-3.0, respectively). Both PU-V2.0 and PU-DA3.0 exhibited a slightly higher amount of char residue (16.8 and 25.0%, respectively) compared to PU-V1.0 (14.1%). We assume that DA-adducts decomposed to initial components via rDA-reaction and thermal stability of BMI and its degradation products was higher than those of furyl-polyurethanes [48] that explains higher chair residue. A similar tendency was observed for the previously synthesized linear PUs: samples with higher BMI content revealed higher char residue [31]. DTG curves of PU-V1.0-3.0 (Figure 3b) also had maximums within the range of 295–310 °C associated with weight loss derived from the thermal decomposition of the furan-containing prepolymer. Small peaks at about 160 °C may be caused by evaporation of DMF traces in the material. The main thermal parameters of the pristine PU-V1.0-3.0 are listed in Table 1. TGA curves for pristine and recycled samples were very similar, proving that recycling does not affect the thermal properties of polymers.

Thermal properties and DA-bonds reversibility of PU-V1.0-3.0 were investigated by DSC (Figure 4). The first heating curves (solid line) of all three polymers demonstrated two endothermic peaks: at 88 and 131 °C for PU-V1.0, 84 and 130 °C for PU-V2.0 and 74 and 130 °C for PU-V3.0 assigned to rDA cleavage of endo and exo isomers, respectively [49]. Endo-adduct as the kinetically favored product has lower rDA-reaction temperature while exo-adducts as thermodynamically-favored products are frequently significantly thermally stable and have higher rDA-reaction temperature [49,50]. DSC-curves of branched PUs were similar to DSC-curves of linear PUs, reported previously: thermograms had two endothermic peaks as well, and the rDA peak temperatures of 115–131 °C for branched materials were not very far from rDA peak temperatures of 124–136 °C for linear PPG-2000-based PUs [31].

Retro Diels–Alder reaction peak was not observed on the second heating curves for all polymer samples (Figure 4, dash lines) that agrees with the complete cleavage of DA-adducts. We also measured Tg values of a soft segment for all the polymeric materials obtained (−54.7 °C, −53.3 °C, and −50.4 °C for PU-V1.0-V3.0, respectively, Appendix A in the Appendix A). We suggest that a slight increase in Tg associated with higher BMI content was due to an increase in overall rigidity and density of the network.

We carried out DSC analysis to demonstrate the DA-bonds formation reversibility after several reprocessing runs using a sample of PU-V3.0 as a representative example (Figure 5). The first heat curves (solid) of all recycled polymer samples (from PU-V3.0 to PU-V3.5) demonstrated rDA-reaction peaks (64–73 °C for endo and 115–131 °C for exo isomer cleavage, respectively), shows successful DA-crosslinks formation after every recycles. Retro DA reaction peak was not observed on the second heating curves for all polymer samples (Figure 5, dash lines). All DSC curves for the pristine and recycled sample of PU-V3.0 were similar in shape and the rDA-peaks were reproduced in a quality manner. 

We also tested all polyurethane samples in the TMA experiment (Figure 6). Thermomechanical analysis in penetration mode was performed for additional investigation of the thermomechanical properties of the obtained materials.

The first transition near −55 °C refers to the soft segment glass transition. The second one (175–184 °C) is the result of the rDA-reaction and dissociation of hard segment hydrogen bonds [51]. The third transition at temperatures higher than 310 °C corresponds to the thermal destruction additionally confirmed with TGA data.

### 3.3. Mechanical Properties and Recycling Tests

Typical loading diagrams for the studied polymers showed that the deformation of polyurethanes depends on the prepolymer/BMI ratio and samples with higher BMI content demonstrated higher Young modulus (Figure 7 and Appendix A in the Appendix A). For PU-V1.0 and PU-V2.0, the samples first deform elastically, i.e., the stress in the material is directly proportional to elongation. With further loading, the materials begin to deform irreversibly and collapse at approximately the same final elongation for both ratios. For polymer with higher BMI content (PU-V3.0) a deformation also starts from the elastic section. Furthermore, a region with small load changes with a noticeable increase in elongation appeared on a curve. This section corresponds to the fluidity of the material. With a further rise in strain for this sample, stress increased due to the orientational hardening of the composition. Moreover, the elongation at break is 50% higher for PU-V3.0 than for materials with a lower prepolymer/BMI ratio. Loading diagrams also show a gentler slope of a linear part of the curve for compositions with lower BMI content.

The difference between diagrams agrees with the higher degree of crosslinking associated with increased BMI content. A denser network results in higher modulus, tensile strength, as well as twice bigger elongation at the break.

We used a dissolution method for the crosslinked PUs reprocessing. Pristine films were cut using hand press to dumbbell-shaped samples for tensile strength tests. After the mechanical experiment, disrupted specimens were cut to even a smaller piece, de-crosslinked, dissolved in DMF at 100 °C, cast in PTFE mold, and cured for 48 h at 60 °C (Figure 8). The same cycle was repeated five times for PU-V1.0-3.0. The stress at break *σ_t_*, the elastic modulus *E*, and the elongation *ε* corresponding to *σ_t_* were calculated from the obtained diagram for virgin samples and four recycles are presented in Figure 9, Figure 10 and Figure 11.

The strength did not change significantly from the number of reprocessing cycles for PU-V1.0 and PU-V2.0 specimens (Figure 9). The tensile strength fluctuated around 8 MPa for the polymer composition PU-V1.0 and 15 MPa for PU-V2.0. In the case of the PU-V3.0, we observed a 50% increase in stress at the break after the first processing. With further processing cycles, the strength of the polyurethane decreased to the original and ceased to depend on the number of processing. The tensile strength, in this case, was about 20 MPa.

For all prepolymer/BMI ratios, the elastic modulus has a maximum at the second processing cycle (Figure 10). Moreover, for PU-V1.2 and PU-V2.2, the module almost doubles compared to the initial values. For PU-V3.2, the maximum during the second processing is less pronounced: the increase in the value of E is comparable with the spread of data. In subsequent processing, the elastic moduli for all compositions return to the level of the original compositions. As noted earlier, the higher the prepolymer/BMI ratio, the higher the elastic modulus. For compositions PU-V1.0-5 and PU-V2.0-5, moduli of elasticity are comparable and differ by no more than 30%. For PU-V3.0-5, the elastic modulus is an order of magnitude higher than for the other two.

In contrast to the tensile strength and elastic modulus, the ultimate elongations of the compositions under tension are nearly independent of the number of processing steps (Figure 11). The level of ultimate elongation depends on the prepolymer/BMI ratio. The elongation at the break fluctuated around 200%, 450%, and 300% for PU-V1.0-1.5, PU-V2.0-2.5, and PU-V3.0-3.5, respectively.

The PU-V1.0-3.0 mechanical properties strongly depend on BMI content. Yong modulus values increased from 13 to 196 MPa (for linear PPG2000-based counterparts from 40 to 98 MPa). Thus, slightly crosslinked branched PUs had more elastic character than linear ones, and at some point, with the growing amount of DA-links, the material became much tougher. Tensile strength values for PU-V1.0-3.0 also show a tendency to grow with BMI content increased (linear materials showed the reverse trend—σ_t_ decreased from 26 MPa for the lowest BMI content to 13 MPa for the highest one).

The morphology of PU-V1.0 and PU-V2.0 was analyzed by transmission electron microscopy. Morphological examination PU-V2.0 samples showed an amorphous phase typical for polyurethanes (Figure 12). Basic units are globules that generate domain structures in particular zones. 

The structural element size distribution (Figure 13) has a bimodal character (especially noticeable in PU-V1.0 sample) due to the difference in polyurethane blocks molecular mass. Domain structures formed from several globules and are not exceeding 300 nm.

The PU-V1.0, and PU-V2.0 samples do not show significant structural differences affected by crosslinks density variation. Certain globules’ size growth is caused by Voranol segments swelling in bismaleimide.

## 4. Conclusions

We designed a new polyurethane-like material that is efficiently recyclable for up to five cycles without loss of strength and heat resistance, based on the reversible dynamic interaction between furan and maleimide. We used furan-rich difurfurylamine to increase the crosslinking density and hence the mechanical properties of the material. We investigated the thermal, mechanical properties, and morphology of materials obtained. We showed that Young modulus strongly depends on BMI content (from 8 to 250 MPa) that allows us to obtain materials of different elasticity as simple as varying BMI content. The reversibility of DA reaction between furan and maleimide components of the material was investigated by NMR spectroscopy, differential scanning calorimetry, and recycle testing. We showed that the stiffness of a material is strictly dependent on the amount of crosslinking agent. Interestingly, stress at the break values and Young modulus has a maximum for all three compositions after the first recycling test, which could be explained by the incompletion of DA-reaction between furane-functionalized prepolymer and maleimide crosslinker in pristine samples. 

## Data Availability

The data presented in this study are available on request from the corresponding author.

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
