# Peer review of "Thermally Remendable Polyurethane Network Cross-Linked via Reversible Diels–Alder Reaction"

_polymers, 2021, doi:10.3390/polym13121935_

Round 1

Reviewer 1 Report

Thermally Remendable Polyurethane Network Cross-Linked 2 via Reversible Diels-Alder Reaction

The authors prepared a series of polyurethane polymers crosslinked with reversible Diels-Alder reaction. The polymers were characterized for their chemical structure and mechanical behavior as well as their ability to be recycled. Below are the revisions I suggest:

  • The name of the TGA equipment should be checked: is it NETZH or NETZCH?
  • Several English language errors should be checked and corrected. For example: Page 4, line 183: prolonger should be changed to prolonged. Page 9, line 284: brake should be changed to break
  • The authors mention fracture strength (figure 9 and in the text) when in reality they have measured stress at break. This should be corrected.
  • The authors use dissolution as a way of recycling. Why was this chosen? This should be justified especially because the polymers are more likely to show excellent recyclability upon re-dissolving. However, in practical applications, thermal reprocessing would be much more suitable.
  • What is the significance of the bimodal distribution of globule size seen in the TEM?
  • What is the novelty of this work? This is not clear and should be highlighted by the authors.

Author Response

We are grateful for the reviewer’s comments that allow us to significantly improve our manuscript. We addressed all remarks point-by-point in this letter

Reviewer #1:

The authors prepared a series of polyurethane polymers crosslinked with reversible Diels-Alder reaction. The polymers were characterized for their chemical structure and mechanical behavior as well as their ability to be recycled. Below are the revisions I suggest:

  1. The name of the TGA equipment should be checked: is it NETZH or NETZCH? 

We agreed with this comment, changes were made.

  1. Several English language errors should be checked and corrected. For example: Page 4, line 183: prolonger should be changed to prolonged. Page 9, line 284: brake should be changed to break

We agreed with this comment, changes were made and we also doublechecked all the writing.

  1. The authors mention fracture strength (figure 9 and in the text) when in reality they have measured stress at break. This should be corrected.

We agreed with this comment, changes were made.

  1. The authors use dissolution as a way of recycling. Why was this chosen? This should be justified especially because the polymers are more likely to show excellent recyclability upon re-dissolving. However, in practical applications, thermal reprocessing would be much more suitable.

The use of thermal reprocessing requires special equipment and optimization of the process, while our goal was to prove the multiple reversibility of the Diels-Alder reaction occurring in the material and compare with the data previously obtained for materials made from linear diols. We agree that for industrial use thermal processing is more convenient, but development of an industrial process was not our goal.

  1. What is the significance of the bimodal distribution of globule size seen in the TEM?

The bimodal distribution is irregular in both samples, which complicates the interpretation of the appearance of extended domain structures. According to the TEM results, it can be seen that the size distribution of structural elements (globules and domains) and the phase structures are very similar for all materials obtained and independent of BMI content.

  1. What is the novelty of this work? This is not clear and should be highlighted by the authors.

As we mentioned in the introduction section “Difurfurylamine, synthesized from furfural, was applied for increasing local furan content and speeding up DA-adduct formation compared to the system using furfuryl alcohol or furfurylamine.”

The Diels-Alder reaction takes place in the solid phase and the mass transfer during interaction is difficult resulting in incompletion of the reaction, especially after recycling. The reaction rate can not be increased by heating due to retro Diels-Alder process that the only way to increase reaction rate is to rise local concentration of reacting partners. Previously furfurylamine or furfuryl alcohol were used as a furan chain extender, which gives a total of two furan fragments per preform molecule. We decided to increase the concentration of furan fragments by using difurfurylamine as a source of furans and thus doubled the concentration of furans up to 4 fragments per preform molecule. We already described the synthesis and study of a number of PUs based on a linear tetrafuran preform. Here we decided to further increase the concentration of furans in the prepolymer, obtain self-healing PUs based on this prepolymer, and study their properties.

Reviewer 2 Report

Presented work is interesting but should be completed in many points:

  • Please, present the reaction scheme in Section 2.2. "Synthesis of difurfurylamine"
  • Please, present the DTG curves for investigated polyurethanes.
  • Parameters from TGA test should be presented in separate table (Td5%, Td10% and Tdmax)
  • The changes in chemical structure (FTIR), glass transition temperature (DSC), thermal stability (TGA) and morphology (TEM) after each reprocessing cycle should be investigated. 
  • For the tensile test, the proper term is "tensile strength" (Figure 9)
  • Please present examplary tensile curves for PU-V1.0-3.0 from test after 1, 2, 3, 4 and 5 cycle of recycling (curves can be presented in supporting informations"
  • Figure 9-11: name of x axis should be recycling cycle
  • Figure 13 should be corrected: some results and text are hidden by axes names
  • The quality of Figures 1-7 and Scheme 1 should be improved

Author Response

We are grateful for the reviewer’s comments that allow us to significantly improve our manuscript. We addressed all remarks point-by-point in this letter.

Reviewer #2: 

Presented work is interesting but should be completed in many points:

  1. Please, present the reaction scheme in Section 2.2. "Synthesis of difurfurylamine".

We agreed with this comment, changes were made.

  1. Please, present the DTG curves for investigated polyurethanes.

We agreed with this comment, changes were made.

  1. Parameters from TGA test should be presented in separate table (Td5%, Td10% and Tdmax).

We agreed with this comment, changes were made.

  1. The changes in chemical structure (FTIR), glass transition temperature (DSC), thermal stability (TGA) and morphology (TEM) after each reprocessing cycle should be investigated.

We performed IR and DSC analysis for each sample after every reprocessing cycle. TGA analysis and morphology test were performed for the pristine samples and after the last reprocessing cycle and they were almost identical so we decided not to test intermediate recycling samples.

  1. For the tensile test, the proper term is "tensile strength" (Figure 9).

We agreed with this comment, changes were made.

  1. Please present examplary tensile curves for PU-V1.0-3.0 from test after 1, 2, 3, 4 and 5 cycle of recycling (curves can be presented in supporting informations"

We add graphics to the supporting information

  1. Figure 9-11: name of x axis should be recycling cycle

We agreed with this comment, changes were made.

  1. Figure 13 should be corrected: some results and text are hidden by axes names

We agreed with this comment, changes were made.

  1. The quality of Figures 1-7 and Scheme 1 should be improved

Changes were made, occasionally word document compressed graphics and now it fixed.

Round 2

Reviewer 1 Report

Good to see the improvements and revisions implemented. Minor English spelling corrections still need to be done. Other than that, I recommend to accept in the present form.

Reviewer 2 Report

Work presents interesting approach for reprocessible cross-linked polyurethanes. Manuscript was improved in terms of reviewer's comments. All completions and corrections are satisfactory. In my opinion it can be accepted in present form.